# The influence of elevated $CO_2$ and soil depth on rhizosphere activity and nutrient availability in a mature *Eucalyptus* woodland

Johanna Pihlblad[1,2], Louise C. Andresen[3], Catriona A. Macdonald[1], David S. Ellsworth[1], Yolima Carrillo[1]

[1] Hawkesbury Institute of environment, Western Sydney University, Penrith Australia
[2] Birmingham Institute for Forest Research, University of Birmingham, Birmingham United Kingdom
[3] Department of Earth Sciences, University of Gothenburg, Gothenburg Sweden

*Correspondence to*: Johanna Pihlblad (m.pihlblad@bham.ac.uk), Louise C. Andresen (louise.andresen@gu.se)

**Abstract.** Elevated carbon dioxide (eCO$_2$) in the atmosphere increases forest biomass productivity, but only where soil nutrients, particularly nitrogen (N) and phosphorus (P) are not limiting growth. eCO$_2$, in turn, can impact rhizosphere nutrient availability. Our current understanding of nutrient cycling under eCO$_2$ is mainly derived from surface soil, leaving mechanisms of the impact of eCO$_2$ on rhizosphere nutrient availability at deeper depths unexplored. To investigate the influence of eCO$_2$ on nutrient availability in soil at depth, we studied various C, N and P pools (extractable, microbial biomass, total soil C and N, and mineral associated P) and nutrient cycling processes (enzyme activity and gross N mineralization) associated with C, N, and P cycling in both bulk and rhizosphere soil at different depths at the Free Air CO$_2$ enrichment facility in a native Australian mature *Eucalyptus* woodland (EucFACE) on a nutrient-poor soil. We found decreasing nutrient availability and gross N mineralization with depth, however this depth associated decrease was reduced under elevated CO$_2$ which we suggest is due to enhanced root influence. Increases in available PO$_4$$^{3-}$, adsorbed P and the C:N and C:P ratio of enzyme activity with depth were observed. We conclude that the influences of roots and of eCO$_2$ can affect available-nutrient pools and processes well beyond the surface soil of a mature forest ecosystem. Our findings indicate a faster recycling of nutrients in the rhizosphere, rather than additional nutrients becoming available through SOM decomposition. If the plant growth response to eCO$_2$ is reduced by the constraints of nutrient limitations, then the current results would call to question the potential for mature tree ecosystems to fix more C as biomass in response to eCO$_2$. Future studies should address how accessible the available nutrients at depth are to deeply rooted plants, and if fast recycling of nutrients is a meaningful contribution to biomass production and the accumulation of soil C in response to eCO$_2$.

## 1 Introduction

With elevated carbon dioxide (eCO$_2$) in the atmosphere, higher photosynthesis rates can drive increases in forest biomass productivity (Ainsworth and Long, 2005; Norby and Zak, 2011). However, enhanced forest productivity in the long-term is not possible in areas where soil nutrients, particularly nitrogen (N) and phosphorus (P) (Fisher et al., 2012) limit growth (Ellsworth et al., 2017; Terrer et al., 2019, 2018). In contrast, plant-microbe interaction under eCO$_2$ might stimulate soil organic matter (SOM) decomposition and alleviate nutrient limitation (Luo et al., 2004; Drake et al., 2011; Wang and Wang, 2021). Higher root exudation rates, stimulation of root growth and fine root production and turnover are all mechanisms that can potentially elicit SOM decomposition and subsequent nutrient release in the rhizosphere (Bernard et al., 2022). Root-mediated changes to SOM decomposition and

nutrient cycling resulting from a changing climate may be especially important in forest systems where tree roots extend far below the soil surface, and where $eCO_2$ may also alter root distribution with depth (Iversen et al., 2008; Iversen, 2010). However, current understanding of nutrient cycling under $eCO_2$ is mainly derived from surface soils, leaving mechanisms of the impact of $eCO_2$ on nutrient availability at deeper depths unexplored (Jackson et al., 1996).

In the organic rich surface layers of soil, where most fine roots are located, microbial activity is high (Graaff et al., 2014). As SOM content, root density, and microbial biomass decline with depth, so does microbial activity and rate of processes in soil (Hobley and Wilson, 2016). Despite this, deeper SOM has been found to be more responsive to fresh C inputs (Fontaine et al., 2007) with the implication that the decomposition effect of fresh C from the rhizosphere is likely to increase with depth. With an extending root system, such as may occur under $eCO_2$ (Iversen, 2010), plants can introduce C where labile C may not have previously been abundant (Iversen et al., 2008; Kuzyakov and Blagodatskaya, 2015) thus promoting microbial activity and accelerated C decomposition at depth, potentially releasing nutrients. Moreover, increased C to the rhizosphere can shift the stoichiometric balance of C relative to soil nutrients (Graaff et al., 2006; Kuzyakov, 2010; Carrillo et al., 2014). With increased abundance of C, the microbial demand for N and P increases (Sistla and Schimel, 2012), in turn leading to an increase in microbial SOM decomposition (Bengtson et al., 2012; Carrillo et al., 2017). Further, microbes have been found to improve their nutrient use efficiency to compensate for the stoichiometric imbalance of decomposer and substrate (Mooshammer et al., 2014). This is manifested through accumulation of N and P in microbial biomass, faster gross mineralization rates, and smaller pools of available inorganic nutrients in the soil solution available for plant uptake. The phenomenon has been found for both N (Rütting et al., 2010) and P (Spohn, 2016; Spohn and Widdig, 2017). How these shifts in stoichiometry manifest in deeper soils is unclear but may have wide ranging implications for forest productivity in response to $eCO_2$.

Belowground allocation of plant-derived C has differential impacts on N and P owing to inherent differences in their cycling. Plant available N in inorganic form (ammonium and nitrate) is derived primarily through SOM decomposition involving the microbial processes of depolymerization and mineralization of organic compounds and through nitrification (Schimel et al., 2015). In contrast, plant available inorganic P (phosphate) can be sourced from both organic sources via microbial SOM decomposition, and inorganic sources via dissolution from primary minerals and desorption from secondary minerals (Adeleke et al., 2017) (Figure 1). Plant and microbial P limitation is often driven by the mechanism of transitioning P between inaccessible organically bound P to an available inorganic form via a dissolved phase, which renders it susceptible to sorption to secondary mineral surfaces like clays and metal hydroxides (Gérard, 2016). In older, highly weathered soils of higher clay content inorganic P availability can be more constraining for plant and microbial activity than N. In these soils, where the primary mineral P source has been depleted, most of the P left in the system is in organic form, either in biomass of plants and microbial cells, or in SOM (Lambers et al., 2008; Walker and Syers, 1976). Increased root exudation and microbial activity in the rhizosphere can increase decomposition of organic P in SOM through phosphatase enzyme production (Bünemann, 2015) and facilitate the release of mineral adsorbed P by releasing organic acids, competing for sorption sites and lowering soil pH. Therefore, the equilibrium of inorganic P between adsorbed and available forms is determined by root exudation, microbial enzyme production and soil mineralogy (Figure 1) all factors that are considered depth-dependent properties.

76         Given that N and P cycling in soil differs, and that the factors controlling those processes can vary with

depth, soil nutrient stoichiometry also tends to vary with depth (Li et al., 2016). Soil C:N ratio tends to decrease
with depth under increased microbial processing of C. Declining SOM content with depth will also lower the N
content. In contrast, soil C:P can decrease, but more often remains unchanged as mineral adsorbed P remains in
soil despite SOM content declining; the potential implication of which, is a reduction in soil N:P at depth (Li et
al., 2016; Zhao et al., 2017). Therefore, many heavily weathered surface soils may be constrained in available
$PO_4^+$, but at depth, some soils may be N limited. This is important in the context of $eCO_2$, because the response
of SOM decomposition to increased labile C availability could be dependent on which nutrient is most limiting to
microbes (Dijkstra et al., 2013), which in turn would be expected to depend on depth. Accordingly, extrapolations
of nutrient limitation from surface soil processes to deeper soil layers become unreliable without accounting for
mechanisms controlling nutrient processing as the stoichiometry changes with depth. The lack of experimental
evidence concerning soil nutrient cycling processes in deeper soil render the assumption that native biomes will
increase their productivity under $eCO_2$ contentious (Iversen et al., 2011; Rumpel and Kögel-Knabner, 2011).

89         The *Eucalyptus* Free Air $CO_2$ Enrichment (EucFACE) facility in eastern Australia has experimentally

exposed a *Eucalyptus* woodland, on a low N and P soil, to $eCO_2$ concentration (+150 ppm) continuously since
2013 (Drake et al., 2016). To date the site has not seen any evidence of increase in aboveground biomass in the
*Eucalyptus* trees under $eCO_2$ (Ellsworth et al., 2017) despite an increase in the photosynthetic rate of both the
dominant tree species and the understory grasses in this ecosystem (Ellsworth et al., 2017; Pathare et al., 2017).
The lack of plant biomass response to the $CO_2$ treatment is hypothesised to be caused by a severe P limitation of
the soil, additions of which was shown to increase plant biomass in a tree stand close by not exposed to $eCO_2$
(Crous et al., 2015). In this system, mineralization and decomposition of SOM have only been investigated in the
upper soil layers (Hasegawa et al., 2016; Castañeda-Gómez et al., 2020, 2021). The potential for the plants in this
system to utilise nutrients in the deeper soil layers of the top meter of soil is relevant because this highly weathered
nutrient poor soil system may already have reached a maximum efficiency for nutrient cycling in the upper soil
layer where SOM and microbial activity is greater. Additionally, Eucalyptus trees are known to have very deep
roots to access water from groundwater aquifers (Laclau et al., 2013), though fine roots capable of nutrient
acquisition are thought to be most abundant in the surface soil layers (Piñeiro et al., 2020). Despite the
considerable number of P limited forests globally there are still large uncertainties surrounding rhizosphere
activity and nutrient cycling in older, P-limited soils compared to younger soils in the northern hemisphere that
are often N limited (Fisher et al., 2012; Terrer et al., 2019).

106         To investigate the influence of $eCO_2$ on nutrient availability in soil at depth, we studied various C, N and

P pools (extractable, microbial biomass, total soil C and N, and mineral associated P) and nutrient cycling
processes (enzyme activity and gross N mineralization) associated with C, N, and P cycling in both bulk and
rhizosphere soil at different depths at the EucFACE facility. We asked: Q1. what is the difference between
rhizosphere and bulk soil in terms of soil properties, and is this changed with soil depth? Q2. what is the effect of
$eCO_2$ on nutrient availability and C:N:P stoichiometry in the rhizosphere, and does it change with soil depth?
Given that increased root exudation will prime microbial nutrient mining, we hypothesize (1) nutrient availability
(inorganic N and P) will be higher in the rhizosphere compared to bulk soil. We also hypothesize that (2) $eCO_2$
will increase availability of P to a greater extent than N in surface soil, but not at deeper layers; and that (3) $eCO_2$
will have less impact on N than P availability and increase the processes contributing to P release (P-targeting
enzymes) more so than N release (N-targeting enzymes and gross N mineralization). This effect will be less
important with depth because the overall N:P ratio declines with depth, alleviating the P limitation and thus
shifting the demand from P to N.

## 2 Materials and methods

### 2.1 Experimental design

The study was performed at the *Eucalyptus* Free-Air $CO_2$ Enrichment (EucFACE) experiment located in a
Cumberland Plain woodland with mature *Eucalyptus* trees in Sydney, Australia (33 37'S and 150 44'E, 23 m
a.s.l.). The site has six experimental rings (n=3), each with a diameter of 25 m. The $CO_2$ treatment was
implemented to three of the rings (e$CO_2$) since September 2012 and reached +150 ppm above ambient $CO_2$ (a$CO_2$)
in February 2013 (Ellsworth et al., 2017). The remaining three rings are controls (a$CO_2$). The soil at the site is a
developing red and/or yellow aeric podsol in weakly organised alluvial deposits (Ross et al., 2020) including iron-
manganese nodules (Clarendon formation) with a metal oxide rich (field observation) transition to a hardpan clay
layer  called Londonderry clay (Atkinson, 1988) found at a variable depth throughout the site (between 35-85
cm). The dominant tree species is *Eucalyptus tereticornis* and the dominant understory grass is *Microlaena*
*stipoides*. The site has an average precipitation of 800 mm per year, with a total precipitation of 16.8 mm in the
month leading up to the sampling campaign. The yearly mean temperature was 17 °C. For further detailed site
description see Ellsworth et al., (2017).

### 2.2 Field sampling, soil preparation and root biomass determination

Soil cores (5 cm diameter) were collected from all rings in September 2017. Twelve cores were taken in each ring,
spread as three in each of the four pre-established two by two-meter subplots designated for soil sampling (4
subplots per ring, total of 72 soil cores). Each core was sampled down to the clay layer which varied with depth
across the site (35-85 cm). Each core was divided into the three depths for investigation: 0-10 cm, 10-30 cm, and
transition (a 10 cm interval where sandy loam transitioned into clay). Samples were kept cool until further
processing in the laboratory within one week of collection. Although the depth of the transition layer differed
throughout the site, the chemical properties are assumed to be similar within this zone across the plots, as the
water periodically builds up above the clay before it drains, creating conditions for podzolification. Soils were
processed to separate bulk from rhizosphere soil. The rhizosphere soil was defined as any soil that was still
attached to the fine roots when these were separated from soil and soil was collected by gently shaking roots. All
other soil in the core was considered bulk soil. For both rhizosphere soil and bulk soil, subplots 1 and 2, and 3 and
4, were combined to two samples per ring and depth (n = 6 samples per ring). This was necessary to have sufficient
rhizosphere soil sample for subsequent analysis. Samples were sieved to < 2 mm. Sub-samples for potential
enzyme activity were frozen (-20 °C) immediately after sieving. Soil samples to be analysed for nutrient
availability and microbial biomass were stored field moist at 5 °C until processed. The roots already handpicked
for rhizosphere soil were washed and dried within a week of sampling and later separated into larger and smaller
than 3mm diameter fractions. Additionally, any remaining roots were hand-picked from a subsample (~50 g) of
sieved soil and scaled to the total sample weight.

**2.3 Extractable carbon, nitrogen, and phosphorus and microbial biomass**

Microbial biomass C, N, and P were determined on fresh soil following the fumigation extraction method of Vance et al. (1987). Briefly, fumigated samples were treated with ethanol free $CHCl_3$ under vacuum (fumigated for four days for C and N, and one day for P) and then extracted for C, N, and P using $K_2SO_4$ and Bray-P I. All extracts were filtered through Whatman 42 grade filter papers and frozen until analysis. Fumigated and unfumigated extracts of $K_2SO_4$ (0.5 M) were analysed for C and N on TOC-L (total organic carbon analyser, Shimadzu corporation, Japan). Fumigated and unfumigated extracts of Bray-P I were analysed for $PO_4^{3-}$, additionally unfumigated $K_2SO_4$ extracts were analysed for inorganic N (ammonium and nitrate), according to Rayment and Lyons (2011), by colorimetry (AQ2 Discrete Analyser, SEAL Analytical, Mequon, WI, USA). Soil was dried (70 °C) for determination of gravimetric soil moisture and air-dried soil was used for pH (1:5 s:w), (S20 SevenEasyTM pH, Mettler-Toledo International Inc., Columbus, OH, USA). Subsamples of the air-dried soil were cleared of visible root fragments and analysed for total soil C and N (LECO TruMac CN-analyser, Leco corporation, USA) and for mineral associated inorganic P.

**2.4 Mineral adsorbed inorganic phosphorus**

To quantify mineral associated inorganic P a one g air-dried subsample was extracted with $NaOH$-$Na_2EDTA$ (0.25M NaOH and 0.05M $Na_2EDTA$) and horizontally shaken for 16 h at 80 rpm after which it was filtered (Rayment and Lyons, 2011). Extracts were diluted 1:10 with sterile water and analysed using the malachite green reagent (Ohno and Zibilske, 1991) in a clear 96 well plate. The plates were analysed by colorimetry on a CLARIOstar plate reader (BMG LABTECH GmbH, Germany) at 610 nm after one hour incubation at 25°C.

**2.5 Pool dilution for gross N mineralization rates**

To assess the gross N mineralization rate an isotope pool dilution assay using $^{15}N$ enriched ammonium was made with a series of laboratory incubations following the method of Rütting et al. (2011). Ammonium concentration and ammonium-$^{15}N$ excess from two time points was done on KCl extracts (Stange et al., 2007; Putz et al., 2018) with SpinMass (Sample Preparation of Inorganic Nitrogen MASSpectrometer) at ISOGOT (Dept of Earth Sciences, University of Gothenburg, Sweden). The $^{15}N$-label was added in duplicate to fresh and sieved soil samples (5 g) with a label consisting of 10 µg ($^{15}NH_4)_2SO_4$ ($^{15}N$ fraction of 99 %, Cambridge Isotope laboratory Inc.) in 0.25 mL milliQ water. After label addition, samples were incubated for 15 minutes and 24 hours under steady temperature (20 °C) and in darkness. The incubations were extracted with 1 M KCl (15 mL), shaken for one hour at 120 rpm and filtered through 42 grade ash-less Whatman filters and frozen until analysis. All gross mineralization rates were calculated using the equation in Kirkham and Bartholomew (1955).

**2.6 Potential enzyme activity method**

Potential activity of seven enzymes associated with C, N and P mineralisation were determined for bulk and rhizosphere soil respectively. For this we used fluorometrically labelled substrates following the method of Bell et al., 2013. Two g frozen soil was mixed to a slurry (1:33 w:v) with MilliQ water in a laboratory blender for one minute. The slurry was pipetted into 96 well plates with three technical replicates and given fluorescent substrates (4-methylumbelliferone; MUB and 7-amino-4-methylcoumarin: MUC) in accordance with the Bell et al. protocol (2013). The samples were then incubated at 25 °C for three hours and analysed for fluorescence with a CLARIOstar plate reader (BMG LABTECH GmbH, Germany). Four enzymes (α-D-glucopyranoside (AG), β-D-

glucopyranoside (BG), β-D-cellobioside (CB), and β-D-xylopyranoside (XYL)) targeted C-rich compounds (sugar, cellulose, hemicellulose), two enzymes (L-Leucine-7-aminopeptidase (LAP) and N-acetyl-β-D-glucosamine (NAG)) targeted N-rich compounds (proteins and chitin), and acid phosphatase (PHOS) targeted organic compounds with P. These enzymes are considered representative of the total enzyme pool active in the soil, however storage in -20 ˚C may have altered the potential enzymatic activity and comparisons with activities in fresh soil from other land-uses should be made with caution (Lane et al., 2022).

**2.7 Statistical analyses**

The impact of $CO_2$ treatment, depth and their interaction were assessed separately for bulk and rhizosphere soil at three depth levels (0-10, 10-30, and transition). Two soil depths (0-10, 10-30) were used in the analysis of rhizosphere where insufficient amounts of rhizosphere soil were recovered during sampling. The subsequent pseudo-replication created with two samples per experimental unit (ring) were dealt with using a linear mixed effects model where $CO_2$ and depth and their interactions were fixed factors and ring a random factor with individual intersects (*lme4* package, Bates et al., 2015), corresponding to the EucFACE experimental design (Hasegawa et al., 2016). To assess the role of $CO_2$ and depth effects on rhizosphere soil, we used a linear mixed effects model with $CO_2$, depth (two depths: 0-10, 10-30) and soil type (bulk, rhizosphere) as fixed factors with all interactions and ring as a random factor with individual intersects (Bates et al., 2015). For gross N mineralization rate in the deepest layer (10 to 30 cm depth) ammonium concentrations in most samples were below detection limit.

Significance was determined with the *ANOVA* function (*car* package, Fox and Weisberg, 2019) with Kenward-Roger degrees of freedom estimation. Post-hoc analysis was performed with the *glht* function for multi-comparison (*multcomp* package, Hothorn et al., 2008). The post-hoc Tukey analysis of all $CO_2$, depth, and soil factors were combined into their unique interactions and then processed in the linear mixed effects model as previously described. Normal distribution of residuals was assessed, and log transformations were performed where required to meet model assumptions.

**3 Results**

**3.1 Fine root biomass**

Fine root biomass density significantly decreased with depth and ranged from 0.12 mg·g$^{-1}$ in the 0-10 cm depth to 2.75 mg·g$^{-1}$ in the transition depth (Figure 2). There was a significant interaction between depth and $CO_2$ where, in the topsoil (0 to 10 cm) elevated $CO_2$ (eCO$_2$) samples had a 28 % lower fine root density than ambient.

**3.2 Carbon in total soil, dissolved and microbial biomass pools**

Dissolved organic carbon (DOC) declined significantly with depth for both bulk and rhizosphere soil, and the decrease by depth was stronger for rhizosphere soil (25 %) than for bulk soil (11 %) (Figure 3A and C). The DOC was significantly higher (by 24 %) in rhizosphere soil than bulk soil (Figure 2 and Table 1) when averaged across depth (0-10 and 10-30 cm depths). Microbial C declined significantly with depth for both bulk soil and rhizosphere soil (Figure 3B and D) and was significantly higher in rhizosphere soil (Table 1, Figure 3) by 36 % (transition was excluded). Total soil C content had a significant effect of depth, and an interaction between $CO_2$ treatment

and depth (Table 1); % soil C content was higher in the 0-10 cm depth under $eCO_2$ but was not different from
ambient in the deeper depths (10-30 and transition) (Table 2).

**3.3 Rate of gross N mineralization and N pools**

Measured soil N content (including $NH_4^+$, $NO_3^-$, microbial N) declined significantly with depth for both bulk and
rhizosphere soils (Figure 4). Ammonium, nitrate, microbial N, and gross N mineralization (Table 1) were
significantly higher in rhizosphere soil than in the bulk soil at both 0 to 10 cm and 10 to 30 cm depths (Table 1).
Total soil N content showed a significant interaction between $CO_2$ treatment and depth (Table 1) where % soil N
content was higher in the 0-10 cm depth under $eCO_2$ but was the same as ambient in the deeper depths (10-30 and
transition).
Gross N mineralization rate declined significantly with depth and was significantly higher in rhizosphere
soil compared to bulk soil; furthermore, $eCO_2$ did not have a significant effect (Figure 5, Table 1). The multiple
comparison showed the 0-10 cm bulk soil samples as being similar magnitude as the rhizosphere 10-30 cm
samples. The 0-10 cm rhizosphere treatment were significantly higher than the ambient 10-30 cm rhizosphere
(Figure 5), though it cannot be statistically separated from any other treatment group due to the high variability.

**3.4 Soil Phosphorus**

The three assessed P contents (extractable $PO_4^{3-}$, microbial P, and mineral associated inorganic P) significantly
declined with increasing depth and were higher in the rhizosphere compared to bulk soil (Table 1 and Figure 6).
For $PO_4^{3-}$ there was a significant interaction between $CO_2$ and depth as the concentration of $PO_4^{3-}$ did not decline
with depth under $eCO_2$. Phosphate concentration in the 10-30 cm depth tended to be higher in $eCO_2$ soils compared
to $aCO_2$ soils (Figure 6D). Microbial P in the bulk soil interacted with $CO_2$ treatment and depth, where microbial
P was lower under $eCO_2$ compared to $aCO_2$ in the 0-10 cm depth only (Figure 6).

**3.5. Enzymatic activity results**

Enzyme activities decreased significantly with depth but did not differ significantly between soil or $CO_2$ treatment
(Table 5 and Table 6). One exception to the general trend was CB (b-D-cellobioside) that did not decrease with
depth and was significantly higher in rhizosphere soil compared to bulk soil. Notable is the difference in
magnitude for N targeting and P targeting enzymes where P enzymes where twice as abundant than N. The two
to one pattern was maintained as the enzyme activity declined with soil depth.

**3.6 Stoichiometry of soil nutrient pools (C, N, P) and soil enzymes**

The C:N and C:P of extractable nutrients in the bulk soil increased significantly with depth by 24.9 and 20.9 units
of C per nutrient, respectively. However, under $eCO_2$ the C:N and C:P stoichiometry did not increase in bulk soil
(Table 3 and 4). The rhizosphere soil N:P ratio significantly declined with depth. When soil was included as an
interactive factor in the model (Table 4), C:N was significant by depth:soil. For extractable C:P ratio both the
interaction between $CO_2$:depth and $CO_2$:soil was significant where C:P ratio declined with $eCO_2$ and depth but
increased with depth when ambient. In the microbial biomass only C:P significantly increased with depth in bulk
soil. The N:P of extractable N and P and microbial biomass stoichiometry significantly increased with depth.
When both bulk and rhizosphere soil was considered (only 0-10 and 10-30 cm depth) soil and depth significantly
affected extractable C:N and N:P, and the interaction of soil and depth was significant for soil C:N (Table 4). The
bulk soil total C:N ratio decreased significantly with depth by 9 units. The rhizosphere soil C:N ratio increased
slightly by only 1 unit, yet still significantly, with depth. There was also a significant interaction between $CO_2$
and depth in the C:N and C:P ratio of the enzymes (Table 5 and Table 6). The C:N and C:P ratios decreased 0.7
and 0.4 units with depth in ambient conditions but increased 0.4 and 0.3 with depth in $eCO_2$. The ratio between N
and P targeting enzymes did not change with depth but was maintained in the range of 0.5-0.7 N enzymes per P
enzyme. The pH showed a marginally significant effect from an interaction of depth and $CO_2$, where the pH
increased slightly in the transition under $eCO_2$ (Table 5).

**4 Discussion**

We sampled rhizosphere soil and bulk soil in a depth profile in a *Eucalyptus* woodland experimentally exposed
to $eCO_2$ for 5 years, with the goal to investigate how root activity influences nutrient availability and stoichiometry
across depth and under $eCO_2$. Supporting our hypothesis (1), the nutrient availability increased in rhizosphere soil
compared to bulk soil. However, we found no clear evidence to support the hypothesis that $eCO_2$ affected the
rhizosphere soil to a greater extent than the bulk soil (Table 1). There was some evidence to support hypothesis
(2), that $eCO_2$ affected the availability of P more than of N as available $PO_4^+$ was more increased with depth in
elevated compared to ambient $CO_2$ (Figure 6). Additionally, the low N:P ratio of enzymes supports hypothesis (3)
that P was more limiting than N (Table 5).

**4.1 Depth effects on soil nutrients and microbial biomass**

The effect of depth was overall significant and the microbial biomass C, N and P, DOC, inorganic N ($NH_4^+$ and
$NO_3^-$), inorganic P ($PO_4^+$), and mineral-adsorbed inorganic P all decreased in availability with depth (Table 1).
However, under $eCO_2$, when bulk and rhizosphere soil were analysed separately the availability of extractable P
in the soil solution in the rhizosphere did not decline with depth (Figure 6D). Increased P availability below
surface soil in the rhizosphere has been found in previous studies at the site (Ochoa-Hueso et al., 2017), which
measured nutrient availability down to 30 cm depth, and in other forest sites investigating nutrient availability in
deeper soil (Blume et al., 2002; Rumpel and Kögel-Knabner, 2011; de Graaff et al., 2014; Li et al., 2016). Notably,
all enzyme activity, including phosphatese activity, declined with depth independatly from $CO_2$ condition (Table
5) indicating that the rhizosphere increase in P availblity in the deeper soil was not due to higher SOM
decomposition. Contrary to the non-response of the microbial C and N concentration,  the microbial P
concentration decreased under $eCO_2$ in the 0-10 cm depth in the bulk soil (Figure 6C), this is similar to the negative
effect of $CO_2$ on fine root density (Figure 2), suggesting that root density and microbial P respond similarly to
$eCO_2$ since both decreased.

293        Stoichiometry changed with depth differently for bulk and rhizosphere soil. The ratio of extractable C to

N and to P in bulk soil increased with depth, as DOC decreased less with depth than inorganic N and P. However,
contrary to our hypothesis the ratio between N and P was constant across depth in bulk soil. Hence, without the
influence of roots, N and P both declined at a similar rate, while the total magnitude of N larger than P as both
decreased with depth. In the rhizosphere soil the ratio between DOC, and inorganic N and P remained constant
with depth while the N:P ratio significantly decreased; hence, the rhizosphere inorganic P became relatively more
available than N at deeper soil. We suggest there was more P available because there were fewer fine roots and
lower microbial biomass to immobilise it. Furthermore, inorganic P decreased with depth more resources were
invested to access it, supported by the consistently higher P targeting enzyme activity than N enzyme activity.

**4.2 Rhizosphere effects on nutrient availability and mineralization across depths**

It is a paradigm in rhizosphere research that microbial activity is high near the root because of the input of energy in the form of newly photosynthesised C (Kuzyakov et al., 2000; Kuzyakov and Cheng, 2001). Supporting this, we found that microbial biomass and nutrient availability was higher in the rhizosphere soil compared to bulk soil. Furthermore, the gross N mineralization rate increased in the rhizosphere compared to bulk soil. Given the positive links found between gross N mineralization and SOM decomposition (Bengtson et al., 2012; Zhu et al., 2014) these findings suggest that root-microbe interactions are facilitating decomposition and increasing nutrient availability (Andresen et al., 2020).

In contrast, the potential activities of enzymes responsible for depolymerizing and hydrolyzing N and P from SOM did not increase closer to the root (Table 5) supporting previous findings from the site that reported enzyme activities were not higher in the presence of roots (Ochoa-Hueso et al., 2017; Castañeda-Gómez et al., 2021). The lack of enzymatic activity response to roots in both surface and deeper soil depths could be due to the microbial community lacking access to energy and N to be able to synthesise enzymes (Olander and Vitousek, 2000), although there is no indication N or C are limiting for enzyme production in this system. Alternatively, because of greater nutrient availability there is reduced need for enzyme production (Sinsabaugh et al., 2009). Finally, a shift in the microbial community composition favouring fungi over bacteria in the rhizosphere as has been observed at the site could lead to lower enzyme production per unit biomass (Castañeda-Gómez et al., 2021).

The stoichiometry of enzymes targeting N and P is an indicator of microbial nutrient demand (Sinsabaugh et al., 2009). In this system, N does not appear to be the most limiting nutrient given the low ratio of N:P targeting enzymes. The low enzyme N:P ratio suggest that P is more highly sought by the microbes in this system (Allison and Vitousek, 2005; Sinsabaugh et al., 2008). We found this independent of soil depth, indicating that P is in higher demand than N in the entire soil profile. Interestingly, no difference in either enzyme amount or stoichiometry was found between bulk soil and rhizosphere soil which indicate that given a higher C availability in the rhizosphere, microbes did not increase their enzyme production to mine for organic P. However, P can also be sourced from non-organic sources (Gérard, 2016). This is supported by the high levels of mineral associated inorganic P in the rhizosphere at depth (Figure 5). We suggest that non-organic sources of P may be important to microbes in the rhizosphere as an alternative to high energy cost enzyme production. Although soil P accumulates in the soil organic fraction with increasing soil age (Crews et al., 1995) this soil is also rich in metal oxides with large surfaces capable of adsorbing phosphate cations (Achat et al., 2016) which root activity in the rhizosphere can release with the help of organic acids without decomposing SOM (Adeleke et al., 2017).

The pattern of decline in nutrient concentrations in deeper soil profiles is well documented (Jobbágy and Jackson, 2001). Though a decline in these concentrations still occurs in the rhizosphere soil with depth, here we can show that root activity counteracts the decline associated with depth, maintaining a higher microbial biomass and nutrient availability in the rhizosphere soil compared to bulk soil (Finzi et al., 2015). Together with the evidence of higher gross N mineralization rate in the rhizosphere soil, we suggest that in this P limited mature forest, roots can drive the availability of both N and P even in deeper soil. Because we did not find a significant increase in potential enzyme activity in the rhizosphere (Table 6) this effect can instead be driven by microbial

biomass turnover, community shift (Castañeda-Gómez et al., 2021) and a strong recycling of nutrients without large decomposition of SOM requiring enzyme activity. Although we can show that deep rhizosphere has an impact on available nutrients our study cannot assess if plants are utilising the increased availability, though increased root turnover has been reported (Piñeiro et al., 2020) suggesting that is the case. However, assuming at least part of plant nutrient immobilisation is via diffusion of concentration gradients (Gilroy and Jones, 2000), a higher nutrient concentration in the deeper rhizosphere soil is likely benefiting plants as well as microbes.

**4.3 Elevated $CO_2$ and depth dependency of rhizosphere effects**

Elevated $CO_2$ increases C availability and nutrients in the rhizosphere through increased rhizodeposition and nutrient mobilisation (Phillips et al., 2011; Kuzyakov et al., 2019). Because root density declines with increasing depth, we hypothesised that the effects of $eCO_2$ on C and nutrient availability will be less important with depth. Contrary to that hypothesis we found that $eCO_2$ interacted with depth by increasing the inorganic P availability at depth under $eCO_2$. Further, mineral associated inorganic P was constantly higher at depth in the bulk soil under $eCO_2$, though the trend is not significant. Metal hydroxide mineral rich clay is capable of strong adsorption of negative ions and organic complexes (Jilling et al., 2018; Rasmussen et al., 2018) which is present at EucFACE. Changes in pH can affect the equilibrium between mineral adsorption and solution concentration though the small increase in pH that was detected in the rhizosphere soil (less than 0.5 units compared to bulk soil, Table 5) is not necessarily enough to change the sorption capacity. Rather the higher $PO_4^{3-}$ adsorption and concentration in solution indicates that higher rates of phosphate processes exist in that space. The different forms of soil P thus appears to respond to different drivers, while the microbial biomass did not immobilise the additionally available $PO_4^{-3}$ or access the mineral associated P. This supports that the microbes are not limited by P at depth. The question remains if plants can access the increased P availability at deeper soils.

The relative content and activity of C-degrading compared to N and P degrading enzymes was higher in the deeper soil under $eCO_2$ for both rhizosphere soil and bulk soil. These trends with depth suggest that the surface soil is more limited by nutrients (i.e. N and P poor) compared to deeper layers where C is a limiting factor for activity. Thus, $eCO_2$ may cause increased microbial activity and enzyme synthesis at depth rather than in the surface soil. The relative content of enzymes for N to P release ranged 0.5 to 0.8, and this indicated biological P limitation rather than N limitation and that ratio was consistent through the depth profile, though the total enzyme activity declined with depth. Only cellulase activity (CB, Table 5) was constant in all layers possibly indicating that plant matter have the potential of being decomposed throughout the soil profile. It was demonstrated by Castaneda-Gomez et al (2020) that root litter decomposition is increased under $eCO_2$ at the site and contributes to C loss from the system. Root litter decomposition can thus be an important source of nutrient release at depth. Further, $eCO_2$ has been found to increase the rate of root turnover in this system (Piñeiro et al., 2020), which is one of the main sources of C supply to the deeper soil, other than increased root exudation.

In this study the observed lack of influence of $eCO_2$ on nutrient availability and N mineralization at the surface is likely due to the topsoil being less limited by C than deeper soils (depth and $CO_2$ interaction). Though enzyme activities decrease with depth, they are more abundant per unit soil C deeper in the profile. Given the rather low $eCO_2$ fertilisation effect found on photosynthetic rate (Ellsworth et al., 2017; Jiang et al., 2020) and root production in this system (Piñeiro et al., 2020) the presumed limited increase in C release belowground is likely turned over without affecting the SOM decomposition. Mineral adsorbed P forms are however sensitive to

root derived changes in pH (Jones and Darrah, 1994), representing a different mechanism for affecting the P cycle separate from SOM decomposition (McGill and Cole, 1981). In the scenario where nutrients mostly become available through recycling, rather than SOM decomposition, it is unlikely that plant nutritional requirements under $eCO_2$ will be satisfied and support continued biomass growth even where roots are known to grow deeper (Iversen et al., 2011). This 'fast-in, fast-out' C cycle in this mature nutrient limited ecosystem under $eCO_2$ will not necessarily release long stored soil C to the atmosphere, but it is not likely to increase C sequestration by gaining additional plant biomass or soil C either. Tough a recent meta-analysis assigning short- and long-term effect of newly fixated C on soil C stocks indicated that any short-term gains of C into SOM could be gone after one to four years (van Groenigen et al., 2017).

There are several consistent trends of an increase in nutrient availability with $eCO_2$ in this study, but they were not statistically significant. These variables include available inorganic N, gross N mineralization rate, inorganic P, and mineral associated P. These trends in pools and processes may indicate an increase in both nutrient availability and up-regulation, if mild, of processes responsible for increased nutrient availability. Though the mature *Eucalyptus* trees have not responded to $eCO_2$ with aboveground biomass growth (Ellsworth et al., 2017) the understory species composition has shifted to include more nutrient-demanding grasses with $eCO_2$ (Hasegawa et al., 2018; Ochoa-Hueso et al., 2021). Higher quality understory litter may in turn drive increased nutrient availability in the soil (Berg and McClaugherty, 1989). Given the necessarily low replication, common to many FACE experiments (Filion et al., 2000), and the lower-than-expected enhancement of photosynthesis in this FACE system (Ellsworth et al., 2017; Pathare et al., 2017; Jiang et al., 2020), an $eCO_2$ effect was expected to be statistically elusive, but here we do show that it can be discerned.

**4.4 Conclusion**

We found that nutrient availability and gross N mineralization were always higher in rhizosphere soil compared to bulk soils, but enzymatic activity was not. The effect of depth, generally, caused a decrease of available nutrients and process rates feeding into the available pools. However, the impact of roots and $eCO_2$ counteracted the decrease found with depth when interactions between soil depth and $CO_2$ or soil depth and soil type (bulk or rhizosphere) occurred. This response of lower concentrations found with increasing depth particularly affected available $PO_4^{3-}$, adsorbed P and the C:N and C:P enzyme activity. We can conclude that roots and $eCO_2$ can affect available nutrient pools and processes well below the surface soil of a forest ecosystem, though it is not clear if the plants can benefit and take up nutrients from deeper parts of the soil profile. Our findings indicate a faster recycling of nutrients in the rhizosphere, rather than additional nutrients becoming available through SOM decomposition. If the tree response to $eCO_2$ is hindered or prevented by nutrient limitations, then the current results would question the potential for mature tree ecosystems to fix more C as biomass in response to $eCO_2$. Future studies are suggested to focus on how accessible the available nutrients at depth are to deeper rooted plants, and if this fast recycling of nutrients is meaningful in production of plant biomass and accumulation of soil C response to $eCO_2$.

**Author contribution**

The initial idea and experimental design were done by Johanna Pihlblad (JP) and Yolima Carrillo (YC) with support by Catriona A. Macdonald (CAM). The data was gathered by JP and with support by YC, CM, and Louise

C. Andresen (LCA). JP did the data management, statistical analysis and wrote the first draft. All other authors contributed to writing of the final paper.

*Code and data availability*: code and data presented in this manuscript can be shared upon request.

*Competing interests*: The authors declare that they have no conflict of interest.

**Acknowledgements**

The authors acknowledge the Dharug nation as the traditional owners of the land on which EucFACE and Western Sydney University is located. We are thankful for support in the field and lab from Vinod Kumar, Craig McNamara, Norbert Klause, Elise Pendall, Jeff Powell, and Laura Castañeda-Gómez. This work was supported by the Australian Research Council Discovery Grant (DP160102452) and the Swedish research council Formas 2017-00423. The EucFACE facility was built as an initiative of the Australian Government as part of the Nation-building Economic Stimulus Package and is supported by the Australian Commonwealth in collaboration with Western Sydney University.

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

**Table 1.** The effect of the factors $CO_2$ (e$CO_2$ and e$CO_2$), soil depth (*0 to 10 cm, 10 to 30 cm, transition*) and soil type (bulk and rhizosphere soil) and their interactions, shown as model F statistic output. Asterisks and bold indicate the level of significance of P values: *** for $P < 0.001$; ** for $P < 0.01$ and * for $P < 0.05$. The extractable nutrients $NH_4^+$, $NO_3^-$ and $PO_4^{3-}$, DOC, and microbial biomass C, N and P are modelled on a $mg \cdot kg^{-1}$ basis, gross N mineralisation rate on a $mg \cdot kg^{-1} \cdot day^{-1}$ basis, and soil C and N in %.

| | $CO_2$ | depth | soil | $CO_2$:depth | $CO_2$:soil | depth:soil | $CO_2$:depth:soil |
|---|---|---|---|---|---|---|---|
| *Df* | 1 | 1 | 1 | 1 | 1 | 1 | 1 |
| **Carbon** | | | | | | | |
| DOC | 0.29 | **30.35** *** | **27.8** *** | 0.01 | 0.05 | 1.46 | 0.94 |
| Microbial C | 0.08 | **141.1** *** | **15.9** *** | 1.92 | 0.6 | 2.34 | 0.01 |
| Soil C | 0.2 | **236.89** *** | 1.21 | 0.1 | **7.94** ** | 0.69 | 1.69 |
| **Nitrogen** | | | | | | | |
| $NH_4^+$ | 0.09 | **24.08** *** | **25.96** *** | 0.27 | 0.2 | 0.03 | 0.16 |
| $NO_3^-$ | 0.46 | **8.96** ** | **16.36** *** | 0.3 | 0.0 | 0.11 | 1.3 |
| Microbial N | 0.16 | **122.42** *** | **18.32** *** | 0.02 | 0.52 | 0.0 | 0.22 |
| gross N min | 2.04 | **13.08** ** | **8.81** ** | 0.37 | 0.05 | 0.92 | NA |
| Soil N | 0.0 | **194.1** *** | 0.19 | 0.01 | **11.04** ** | 2.68 | 2.42 |
| **Phosphorus** | | | | | | | |
| $PO_4^{3-}$ | 0.37 | **32.63** *** | **33.18** *** | **8.6** ** | 2.21 | 0.17 | 0.06 |
| Microbial P | 0.48 | **126.46** *** | **6.38** * | 2.53 | 0.0 | 0.18 | 0.11 |
| Mineral Pi *a* | 0.03 | **68.31** *** | **5.77** ** | 0.19 | 0.58 | 2.34 | 0.73 |

**Table 2.** Total soil C and N (%) and the C to N ratio for ambient a$CO_2$ and elevated e$CO_2$ in bulk soil at the three depths. Standard error is given in parenthesis. Results from statistical analysis are provided in Table 1.

| Depth | Soil C % Ambient | Soil C % Elevated | Soil N % Ambient | Soil N % Elevated | C:N Ambient | C:N Elevated |
|---|---|---|---|---|---|---|
| *0-10* | 1.46 (0.2) | 1.83 (0.2) | 0.09 (0.0) | 0.11 (0.0) | 15.86 (0.6) | 16.05 (0.4) |
| *10-30* | 0.52 (0.1) | 0.59 (0.1) | 0.04 (0.0) | 0.05 (0.0) | 12 (1.1) | 12.37 (0.9) |
| *transition* | 0.15 (0.0) | 0.17 (0.0) | 0.02 (0.0) | 0.02 (0.0) | 6.59 (1.1) | 7.34 (1.0) |

686

687

**Table 3.** Extractable and microbial C, N and P stoichiometry (mg kg$^{-1}$/mg kg$^{-1}$) and soil C:N ratio for bulk soil (B) on the left of each column and rhizosphere soil (R) on the right for a mature Eucalyptus forest soil exposed to ambient and elevated $CO_2$ for three depths (0 to 10 cm, 10 to 30 cm, transition). Stoichiometry was calculated on a mg kg$^{-1}$ mass basis with standard error below in parenthesis.

| | Extractable C:N B | Extractable C:N R | Extractable C:P B | Extractable C:P R | Extractable N:P B | Extractable N:P R | Microbial C:N B | Microbial C:N R | Microbial C:P B | Microbial C:P R | Microbial N:P B | Microbial N:P R | Soil C:N B | Soil C:N R |
|---|---|---|---|---|---|---|---|---|---|---|---|---|---|---|
| **Ambient** | | | | | | | | | | | | | | |
| *0-10* | 17.9 (3.5) | 15.0 (3.5) | 24.2 (1.6) | 29.9 (3.1) | 1.6 (0.2) | 2.5 (0.5) | 5.6 (0.6) | 5.1 (0.4) | 9.8 (0.6) | 11.0 (0.9) | 2.0 (0.2) | 2.2 (0.1) | 15.9 (0.6) | 14.6 (0.5) |
| *10-30* | 36.3 (10.4) | 19.2 (5) | 30.5 (3.2) | 31.3 (6.4) | 1.5 (0.5) | 1.9 (0.3) | 4.3 (0.6) | 4.4 (0.8) | 12.3 (2) | 13.4 (3.5) | 2.7 (0.2) | 2.9 (0.3) | 12.0 (1.1) | 15.6 (0.8) |
| *transition* | 56.8 (22.2) | NA | 65 (18.7) | NA | 1.9 (0.7) | NA | NA | NA | NA | NA | 9.5 (3) | NA | 6.6 (1.1) | NA |
| **Elevated** | | | | | | | | | | | | | | |
| *0-10* | 14 (2.6) | 12.6 (2.4) | 26.1 (2.4) | 25.3 (0.2) | 2 (0.1) | 2.3 (0.4) | 5.8 (0.3) | 6.2 (1.1) | 12.5 (1.2) | 20.1 (7.3) | 2.2 (0.2) | 3.3 (1.3) | 16.0 (0.4) | 16.0 (0.5) |
| *10-30* | 20.5 (6.3) | 12.9 (2.5) | 23.4 (1.4) | 22.1 (1.8) | 1.5 (0.3) | 2.0 (0.3) | 7.5 (1) | 5.5 (0.9) | 14.9 (2.8) | 16.8 (3.1) | 2.3 (0.2) | 2.7 (0.3) | 12.4 (0.9) | 16.9 (0.9) |
| *transition* | 24.3 (8.5) | NA | 24.8 (7) | NA | 1.2 (0.3) | NA | 8.3 (5) | NA | NA | NA | 17.8 (13.6) | NA | 7.3 (1) | NA |

692

693

**Table 4.** Model F statistic and significance of extractable and microbial C, N and P, and soil C:N. Where bulk and rhizosphere are shown separate, bulk was modelled with 3 depth levels whereas rhizosphere soil was modelled with only 2. Where bulk soil and rhizosphere soil are shown together (†) only the 0-10 and 10-30 cm depths are included in the model. Significance of P values are as indicated: *** indicate $P < 0.001$; ** indicate $P < 0.01$ and * indicates $P < 0.05$.

| | Extractable C:N | Extractable C:P | Extractable N:P | | Microbial C:N | Microbial C:P | Microbial N:P | Soil C:N |
|---|---|---|---|---|---|---|---|---|
| **Bulk** | | | | | | | | |
| $CO_2$ | 0.32 | 0.62 | 0.04 | | 0.45 | 0.16 | 0.3 | 0.16 |

|  | Extractable | | | Microbial | | | Soil |
|---|---|---|---|---|---|---|---|
|  | C:N | C:P | N:P | C:N | C:P | N:P | C:N |
| depth | **4.8 *** | 0.51 | 1.7 | 0.67 | 0.78 | **11 *** ** | **62.4 *** ** |
| $CO_2$:depth | 0.34 | 2.48 | 0.84 | 0.62 | 0.12 | 1.27 | 0.06 |
| **Rhizosphere** | | | | | | | |
| $CO_2$ | 0.14 | 0.77 | 0.01 | 0.62 | 0.54 | 0.23 | 3.9 |
| depth | 0.46 | 1.6 | 2.01 | 1.97 | 0.02 | 0.8 | 1.91 |
| $CO_2$:depth | 0.36 | 0.6 | 0.04 | 0.45 | 0 | 0.42 | 0 |
| **Bulk and Rhizosphere†** | | | | | | | |
| $CO_2$ | 0.21 | 0.55 | 0.07 | 0.84 | 0.3 | 0.08 | 2.02 |
| depth | **6.93 *** | 0 | **7.91 **** | 1.16 | 0.27 | 2.5 | **9.27 **** |
| soil | **11.8 **** | 0.06 | **13.58*** ** | 1.73 | 1.28 | 1.53 | **7.4 *** |
| $CO_2$:depth | 0.52 | 3.23 | 0.06 | 1.57 | 0.02 | 1.54 | 0.01 |
| $CO_2$:soil | 0.2 | 1.78 | 1.35 | 0.04 | 1.47 | 0.29 | 0.96 |
| depth:soil | 3.04 | 3.01 | 0.84 | 0.94 | 0.13 | 0.04 | **19.12 *** ** |
| $CO_2$:depth:soil | 0.01 | 0.58 | 0.27 | 0.2 | 0 | 0.03 | 0.01 |

699

**Table 5.** Potential enzyme activity and stoichiometry of enzymes targeting C, N and P compounds (µmol h-1 g$^{-1}$) for bulk and rhizosphere soil of a mature Eucalyptus forest soil exposed to ambient and elevated $CO_2$ for three depths (0 to 10 cm, 10 to 30 cm, transition), with standard error in parenthesis. Four enzymes (α-D-glucopyranoside (AG), β-D-glucopyranoside (BG), β-D-cellobioside (CB), and β-D-xylopyranoside (XYL)) targeted C-rich compounds (sugar, cellulose, hemicellulose), two enzymes (L-Leucine-7-aminopeptidase (LAP) and N-acetyl-β-D-glucosamine (NAG)) targeted N-rich compounds (proteins and chitin), and acid phosphatase (PHOS) targeted organic compounds with P.

| | Enzyme | | | | | | | Sum | | | Stoichiometry | | | |
|---|---|---|---|---|---|---|---|---|---|---|---|---|---|---|
| Layer | AG | BG | CB | XYL | LAP | NAG | PHOS | C | N | P | C:N | C:P | N:P | pH |
| **Bulk Ambient** | | | | | | | | | | | | | | |
| *0-10* | 5.3 (1) | 38.9 (7.9) | 16.4 (3.3) | 23.5 (5.1) | 33.8 (11.5) | 32.1 (5.3) | 121.9 (27.3) | 84 (14.1) | 65.9 (12.5) | 121.9 (27.3) | 1.5 (0.3) | 0.8 (0.1) | 0.7 (0.2) | 5.8 (0.1) |
| *10-30* | 3.5 (1) | 9.5 (1.7) | 4.1 (1) | 6.6 (1.2) | 16.3 (4.3) | 10.4 (0.8) | 47.6 (10.2) | 23.6 (4) | 26.8 (4.6) | 47.6 (10.2) | 1.2 (0.4) | 0.8 (0.3) | 0.6 (0) | 6 (0.1) |
| *transition* | 1.6 (0.6) | 2.5 (1) | 1.1 (0.4) | 1.4 (0.5) | 9.3 (1.8) | 5.2 (1.4) | 25.0 (6.3) | 6.6 (2.3) | 14.5 (2.7) | 25.0 (6.3) | 0.7 (0.3) | 0.3 (0.1) | 0.7 (0.2) | 5.8 (0.1) |
| **Bulk Elevated** | | | | | | | | | | | | | | |
| *0-10* | 5.3 (1.3) | 35.8 (11.3) | 12.5 (3.9) | 20.9 (6.7) | 23.8 (7.5) | 31.7 (10.1) | 139.5 (52) | 74.5 (22.3) | 55.5 (15.4) | 139.5 (52) | 1.4 (0.2) | 0.7 (0.2) | 0.5 (0.1) | 5.7 (0.2) |
| *10-30* | 5.8 (1.6) | 15.4 (5.7) | 6.9 (2) | 11.1 (2.7) | 13.7 (3.3) | 17 (4) | 65.9 (18) | 39.2 (10.5) | 30.7 (5.8) | 65.9 (18) | 1.4 (0.3) | 0.8 (0.3) | 0.6 (0.1) | 5.9 (0.1) |
| *transition* | 4.6 (1.3) | 7.3 (1.8) | 4.7 (1.2) | 5.2 (1.2) | 3.4 (1.1) | 16.1 (9.3) | 23.6 (5.2) | 21.7 (4.5) | 19.5 (10.1) | 23.6 (5.2) | 2 (0.5) | 1.1 (0.3) | 0.7 (0.2) | 6.1 (0.2) |
| **Rhizosphere Ambient** | | | | | | | | | | | | | | |
| *0-10* | 5.2 (1.7) | 52.4 (17.7) | 16.3 (3.1) | 21.8 (6.6) | 33.6 (13.4) | 35.6 (9) | 119.9 (33.4) | 95.7 (26.8) | 69.2 (14.1) | 119.9 (33.4) | 1.6 (0.4) | 0.8 (0.1) | 0.7 (0.2) | 5.9 (0.1) |
| *10-30* | 5.3 (1.3) | 12.5 (1.4) | 7.7 (1.6) | 9.9 (1.3) | 16.5 (4.4) | 13.5 (1.8) | 61.4 (13) | 35.5 (4.4) | 30 (4.9) | 61.4 (13) | 1.4 (0.3) | 0.9 (0.3) | 0.5 (0.1) | 5.9 (0.1) |
| *transition* | 4.3 (1.6) | 12.3 (6.1) | 6.5 (3.4) | 9.4 (4.1) | 13.3 (2.4) | 19.7 (10.2) | 56.2 (13.9) | 32.4 (14.5) | 33 (11.6) | 56.2 (13.9) | 1 (0.3) | 0.5 (0.1) | 0.6 (0.1) | 5.7 (0.1) |

| | Enzyme | | | | | | | Sum | | | Stoichiometry | | | |
|---|---|---|---|---|---|---|---|---|---|---|---|---|---|---|
| **Layer** | **AG** | **BG** | **CB** | **XYL** | **LAP** | **NAG** | **PHOS** | **C** | **N** | **P** | **C:N** | **C:P** | **N:P** | **pH** |
| **Rhizosphere Elevated** | | | | | | | | | | | | | | |
| *0-10* | 3.9 (1.2) | 34.4 (8.1) | 12.4 (3.5) | 20.1 (4.3) | 25.1 (7.4) | 29.7 (6.9) | 126.1 (40.6) | 70.8 (16.3) | 54.8 (12.9) | 126.1 (40.6) | 1.3 (0.1) | 0.7 (0.1) | 0.5 (0.1) | 5.7 (0.2) |
| *10-30* | 6.6 (2.1) | 17.8 (3.2) | 6.8 (1) | 11.4 (1.4) | 16 (2.6) | 23.9 (4) | 97.1 (24.6) | 42.6 (4.3) | 40 (5.7) | 97.1 (24.6) | 1.2 (0.2) | 0.7 (0.2) | 0.5 (0.1) | 5.8 (0.1) |
| *transition* | 4.5 (1.3) | 17.2 (3.8) | 10.4 (3.5) | 6.3 (1.5) | 5.4 (1.1) | 32.1 (15.5) | 53.1 (16.8) | 38.3 (5.2) | 37.5 (15.8) | 53.1 (16.8) | 1.4 (0.3) | 0.9 (0.2) | 0.8 (0.3) | 6 (0.3) |

**Table 6:** Model F statistic and significance levels for potential enzyme activity. Significance of P values are in bold and as indicated: *** indicate $P < 0.001$; ** indicate $P < 0.01$ and * indicates $P < 0.05$.

| | | | | | | | | sum | | | stoichiometry | | | |
|---|---|---|---|---|---|---|---|---|---|---|---|---|---|---|
| | **AG** | **BG** | **CB** | **XYL** | **LAP** | **NAG** | **PHOS** | **C** | **N** | **P** | **C:N** | **C:P** | **N:P** | **pH** |
| $CO_2$ | 0.98 | 0 | 0.01 | 0.03 | 0.8 | 1.55 | 0.19 | 0.02 | 0 | 0.19 | 1.53 | 0.72 | 0.14 | 0.03 |
| depth | 1.45 | **23.28** *** | **18.44** *** | **22.84** *** | **11.96** *** | **6.37** ** | **17.62** *** | **24.2** *** | **14.41** *** | **17.62** *** | 0.51 | 0.48 | 0.73 | 0.67 |
| soil | 0.9 | 2.42 | **3.05** (.) | 0.83 | 0.22 | 2.59 | 1.48 | 2.43 | 2.03 | 1.48 | 0 | 0 | 0 | 0.17 |
| $CO_2$:depth | 1.25 | 1.77 | **2.81** (.) | 0.57 | 0.42 | 1.16 | 0.42 | 1.83 | 1.13 | 0.42 | **3.3** * | **4.42** * | 1.03 | **2.94** (.) |
| $CO_2$:soil | 1.01 | 0.38 | 0.15 | 0.43 | 0.01 | 0 | 0.01 | 0.51 | 0 | 0.01 | 1.84 | 1.13 | 0.04 | 0.04 |
| depth:soil | 1.02 | 0.27 | 1.56 | 0.81 | 0.06 | 1.01 | 0.96 | 0.59 | 0.74 | 0.96 | 0.03 | 0 | 0.05 | 0.38 |
| $CO_2$:depth:soil | 0.07 | 0.41 | 0.29 | 0.25 | 0.02 | 0.12 | 0.12 | 0.04 | 0.06 | 0.12 | 0.34 | 0.22 | 0.07 | 0 |

**Figures**

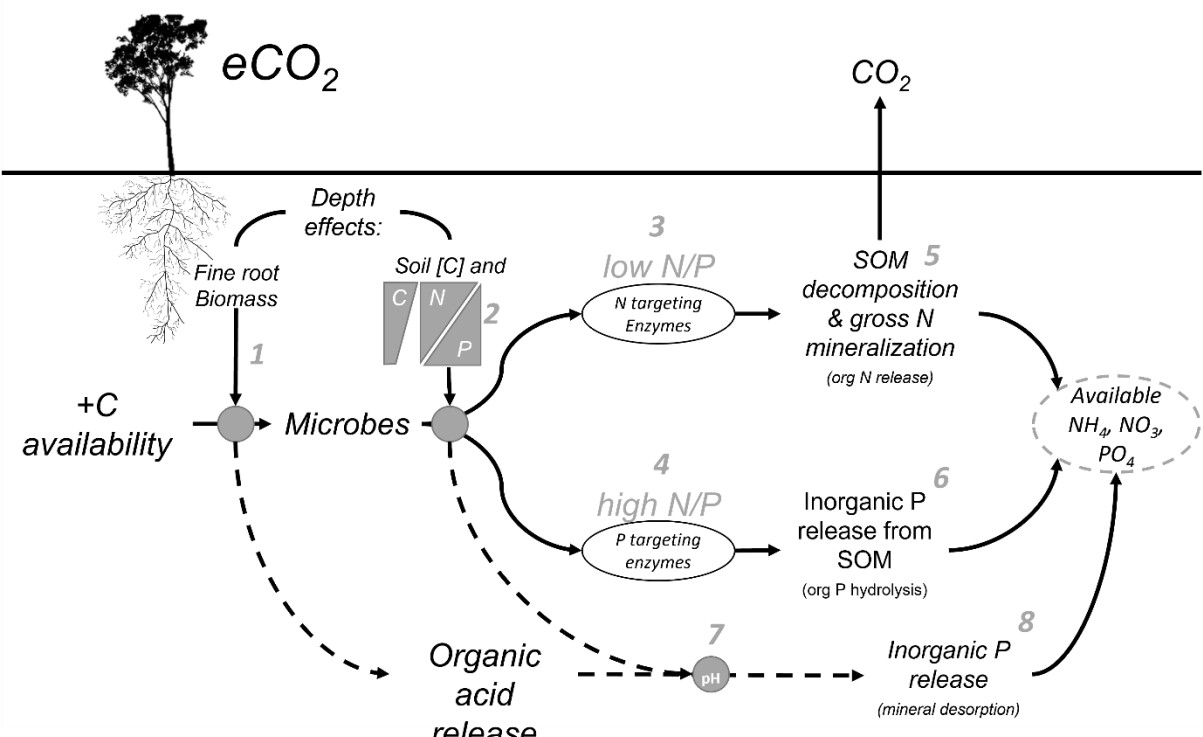


**Figure 1.** Conceptual diagram of the mechanisms affecting nutrient availability as influenced by soil depth.
Elevated $CO_2$ increases C availability belowground, but the effect of that extra C is moderated by depth dependent
mechanisms. (1) Root exudation in the rhizosphere soil is proportional to fine root biomass which decreases with
depth. (2) The microbial strategy to release nutrients is a function of soil C content and N to P ratio, which also
can change with depth. (3) The microbial strategy is a response to the N to P ratio either producing N targeting
enzymes in low N to P conditions or (4) P targeting enzymes in high N to P conditions. (5) Nitrogen targeting
enzymes act to decompose SOM and increase gross N mineralization, transforming N into $NH_4^+$ and ultimately
$NO_3^-$ which are available for plant uptake. (6) P targeting enzymes cut phosphates from organic molecules by
hydrolysis. (7) One further mechanism behind nutrients release affected by $eCO_2$, is that soil pH is changed,
impacting the soil sorption capacity, by the organic acid exudates from roots and microbial mineralization thereof.
(8) The decreased acidity tips the balance of phosphates in solid and in solution, to increase soil solution content
and P availability by mineral desorption.

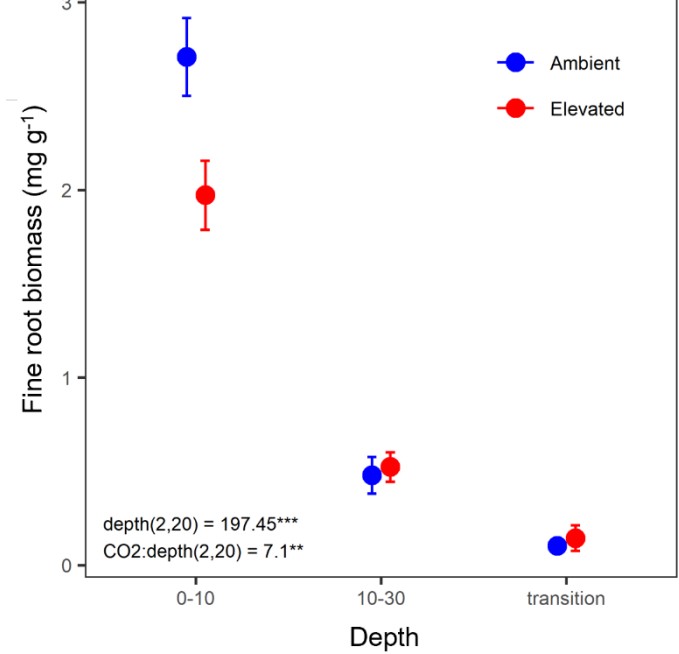


**Figure 2.** Biomass of fine roots of less than 3 mm thickness (mg·g⁻¹) in the mature *Eucalyptus* forest soil exposed
to ambient (blue) and elevated (red) $CO_2$ for three depths (0-10 cm, 10-30 cm, transition). Error bars indicate
standard error. Mixed effects model output stated with (degrees of freedom, Df residuals) F statistic presented and
asterisks for the P values for significance are as indicated: *** indicate $P < 0.001$ and ** indicate $P < 0.01$.

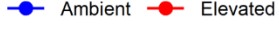

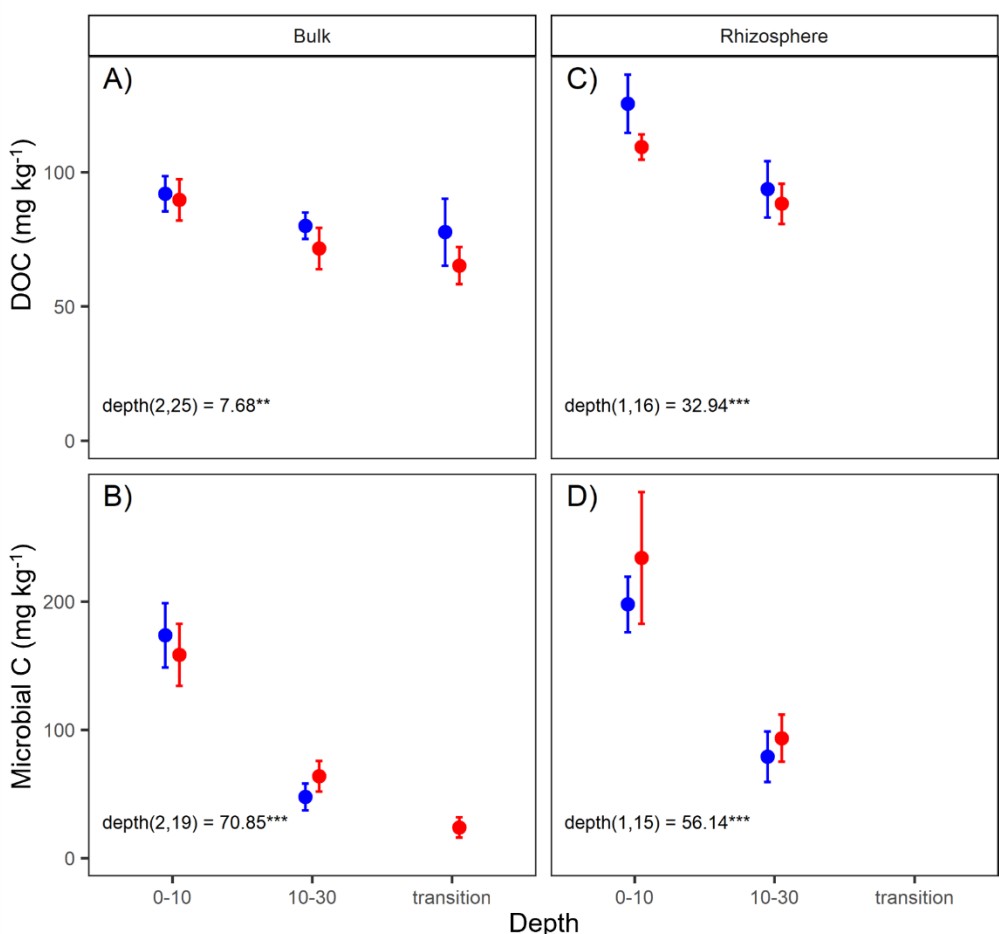


**Figure 3.** Dissolved organic carbon (DOC) and microbial biomass carbon (C) content for bulk and rhizosphere
soil of the mature *Eucalyptus* forest soil exposed to ambient (blue) and elevated (red) $CO_2$ for three depths (0 to
10 cm, 10 to 30 cm, transition). Error bars indicate standard error. Mixed effects model output stated with (degrees
of freedom, Df residuals) and F statistic presented and asterisks for the P values for significance are as indicated:
*** indicate P < 0.001 and ** indicate P < 0.01. Results from statistical analysis of comparison of soil types (bulk
and rhizosphere) are presented in Table 1.

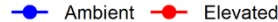

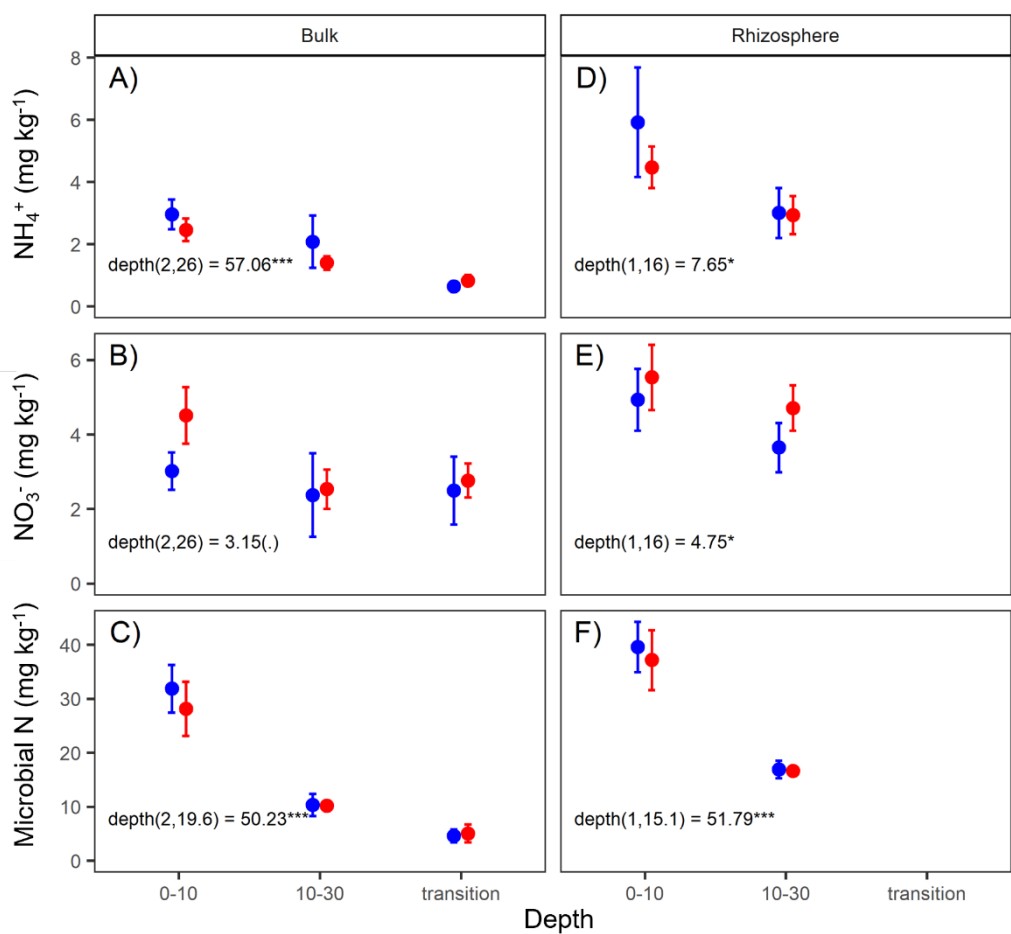


**Figure 4.** Nitrogen (N) pools in the forms of ammonium ($NH_4^+$), nitrate ($NO_3^-$) and microbial biomass N for bulk and rhizosphere soil of the mature *Eucalyptus* forest soil exposed to ambient (blue) and elevated (red) $CO_2$ for three depths (0 to 10 cm, 10 to 30 cm, transition). Error bars indicate standard error. Mixed effects model output stated with (degrees of freedom, Df residuals) and F statistic presented and asterisks for the P values for significance are as indicated: *** indicate $P < 0.001$, ** indicate $P < 0.01$, * indicates $P < 0.05$ and (.) indicates a tendency to a significance $P < 0.1$. Results from statistical analysis of comparison of soil types (bulk and rhizosphere) are presented in Table 1.


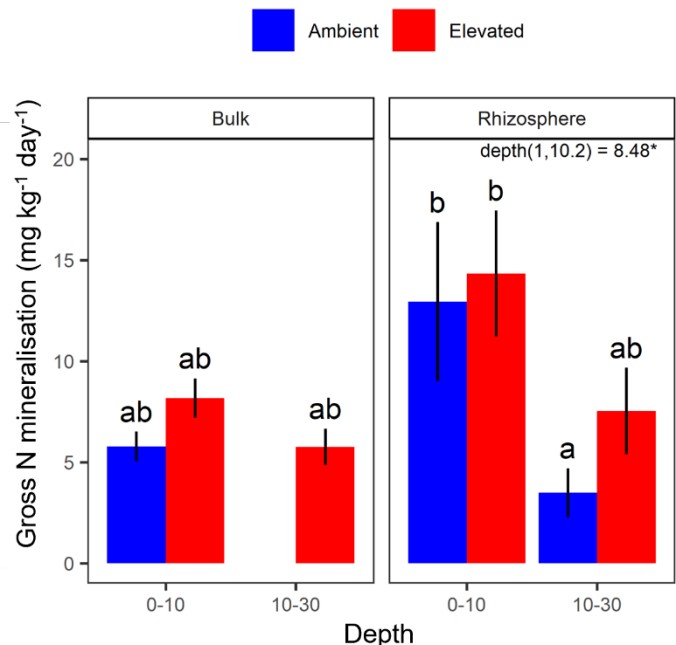


**Figure 5.** Gross N mineralization for bulk and rhizosphere soil of the mature *Eucalyptus* forest soil exposed to
ambient (blue) and elevated (red) $CO_2$ for two depths (0 to 10 cm, 10 to 30 cm). Error bars indicate standard error.
Mixed effects model output stated with (degrees of freedom, Df residuals) and F statistic presented and asterisks
for the P value for significance, * indicates $P < 0.05$. Results from statistical analysis of comparison of soil types
(bulk and rhizosphere) are presented in Table 1.

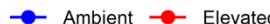

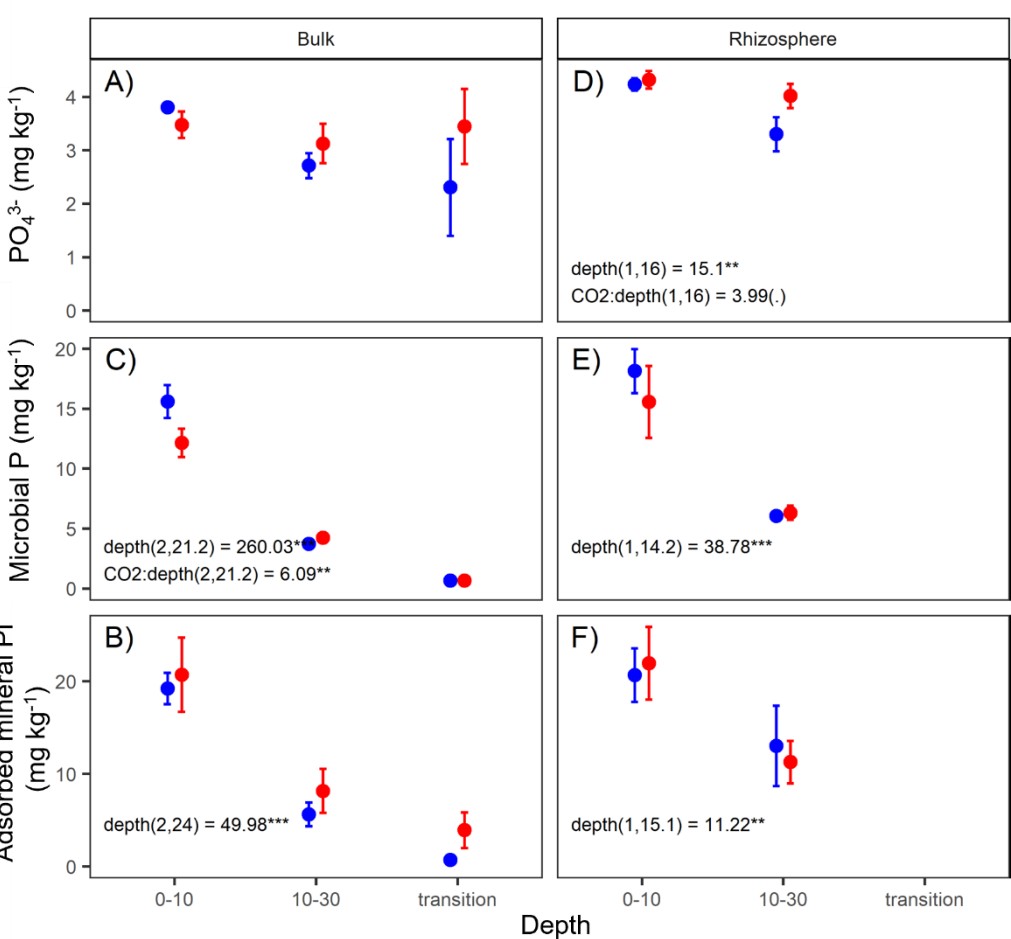


**Figure 6.** Measured soil P pools in the in forms of inorganic P (PO$_4^{3-}$), microbial biomass P, and mineral associated phosphate through adsorption for bulk and rhizosphere soil of the mature *Eucalyptus* forest soil exposed to ambient (blue) and elevated (red) CO$_2$ for three depths (0 to 10 cm, 10 to 30 cm, transition). Error bars indicate standard error. Mixed effects model output stated with (degrees of freedom, Df residuals) and F statistic presented and asterisks for the P values for significance are as indicated: *** indicate P < 0.001, ** indicate P < 0.01, * indicates P < 0.05 and (.) indicates a tendency to a significance P < 0.1. Results from statistical analysis of comparison of soil types (bulk and rhizosphere) are presented in Table 1.