# Peer review of "The influence of elevated CO2 and soil depth on rhizosphere activity and nutrient availability in a mature *Eucalyptus* woodland"

_Biogeosciences, 2022_

## Author Response (AR1)

**Dear Associated Editor,**

We are thankful for the opportunity to submit the suggested changes by reviewer #1 and #2. We have revised the manuscript per their suggestions and the notes given by the associated editor on some minor formatting issues with the original submission. Please find the manuscript The influence of elevated $CO_2$ and soil depth on rhizosphere activity and nutrient availability in a mature Eucalyptus woodland" by Johanna Pihlblad et al., Biogeosciences Discuss (bg-2022-145) enclosed as well as specific replies to each reviewer and how the edits were made.

Kind regards on behalf of all listed authors,

Dr Johanna Pihlblad

**Dear Reviewer #1**

The authors would like to thank the anonymous referee #1 for the thorough attention and skill with which the referee read and suggested edits as well as general comments to the manuscript titled "The influence of elevated CO2 and soil depth on rhizosphere activity and nutrient availability in a mature Eucalyptus woodland" by Johanna Pihlblad et al., Biogeosciences Discuss (bg-2022-145). We are confident that the edits and comments suggested by anonymous reviewer #1 has made the manuscript into a more clear and higher quality manuscript.

Briefly we decided to adhere to the reviewers' main general comment to strengthen the link to previous results as well as introduce the enzyme data in the main text along with their methodology. However, though we agree with the reviewer #1 that the transition layer is not the main discussion point of the paper, we do think there is value in presenting the transition layer data and statistical testing of the transition layer data in the main text given it is not a common endeavor to look at the deeper layers of soil (specifically clay related properties at depth) and the microbial usage of nutrients in this sphere under elevated $CO_2$ or indeed in any large-scale ecological experiments. Though the knowledge gained is based on negative results the novel information is of value for further studies informing on an area of soil affected by climate change factors that have previously not been included in the scientific cannon. For this reason, we want to keep it in the main text though we deeply appreciate the attention to detail and skill by which the reviewer has considered the manuscript as a whole.

Below we have detailed the changes made to the manuscript and if when not changed as suggested the reasons for not doing so.

Kind regards on behalf of all listed authors,

Dr Johanna Pihlblad

**Specific comments**

Lines 142-143: Freezing the soil samples prior to enzyme analysis is not ideal, as freezing and thawing could kill the microbes. Why was this approach chosen? I recognize that this cannot be changed for the publication, but I recommend the authors to write a few sentences about the caveats of this and perhaps only discuss their results in terms of the differences between treatments, since the absolute values could be well underestimated. Another option is to make a quick comparison with other studies, perhaps in similar areas/conditions, to show that the absolute values were not affected by the soil preparation method chosen.

Suggested references for the debate:

https://www.sciencedirect.com/science/article/pii/S0038071712004476?casa_token=_VQgFC080-sAAAAA:0OipU3cTduPiXo0_g6QWb8HCel8rb2lO_jxEYQR6uWx_tIg38yq4T_IS8IxXmeqw4vbbRdSoMg

https://www.sciencedirect.com/science/article/pii/S0038071709004441?casa_token=yD92bxS-vEQAAAAA:c3l8X3LT_9bRJ84K2fZ6g5P0P3E1abvweiZlv0u_ok17lYxNs4eix5oZw9vAombmMqdVl_Vnpg

Why not include the enzyme methodology into the main manuscript document?

> The reason for freezing soil after harvesting prior to the analysis of potential enzyme activity was mainly due to two reasons; firstly, the time restrictions during the field harvest, making sure the soil

was processed no longer than 7 days was crucial to maintain the validity not just for the enzyme analysis but also for the other microbial pools and rates of transformation as well as C and nutrient pools. Collecting the field samples was done over 5 days followed by a day of sieving and processing leaving no time to mauver around analysing potential enzyme activity on fresh soil. Secondly, to not cause the enzymes to break down due to hot temperatures often experienced on the eastern coast of NSW, Australia. If the samples were kept fresh or air dried and stored for the analysis the sometimes high temperatures experienced in this region can degrade the enzymes faster than in the milder temperate or Mediterranean climates of the northern hemisphere. To minimize temperature degradation of the soil enzymes, which can happen in hotter climates, aliquots of soils were frozen in -20 ˚C. Other studies from the same region have all used frozen soils for their potential enzyme activity including but not limited to:

Hasegawa, S., Macdonald, C.A. and Power, S.A. (2016), Elevated carbon dioxide increases soil nitrogen and phosphorus availability in a phosphorus-limited *Eucalyptus* woodland. Glob Change Biol, 22: 1628-1643. https://doi-org.ezproxy.uws.edu.au/10.1111/gcb.13147

Additionally, a standard dry sample was included on every plate prepared for analysis allowing for inter plate comparison. No meaningful difference outside of the expected variability was observed, though the trend for the air-dried sample to have lower activity was note (data not shown). Also, of note here is that the sample size for this comparison was small.

The actions taken in the manuscript are as follows: the enzyme data table and methodology were moved from the supplement to the main body of text as suggested by the reviewer and the enzyme data were integrated more though-out the discussion section. Further, the caveats for the use of frozen soil was introduced in the method section 2.6 stating: "…however storage in -20 ˚C may have altered the potential enzymatic activity and comparisons with activities in fresh soil from other land-uses should be made with caution (Lane et al., 2022)."  (line 194)

Line 188: How did you deal with it? Data was transformed to log as described in lines 194-195?

It's not uncommon for data describing gross rates to miss values and have skewed distributions between treatments, for this reason we are removing the current description and instead writing the following for clarity: "For gross N mineralization rate in the deepest layer (10 to 30 cm depth) ammonium concentrations in most samples were below detection limit." (line 205)

Lines 273-274: It seems that this sentence is contradicting the previous ones in this paragraph. If there was more P available with depth (because you argue there are less roots and microbial activity in those deeper layers), why do you state that "P became limiting at depth"? I would understand that overall P is more limiting than N, as supported by your enzyme results, but the depth argument is not very clear to me in this section. Could you clarify this, please?

The section was clarified by adding the sentence: "Hence, without the influence of roots, N and P both declined at a similar rate, while keeping the total magnitude of N larger than P as both decreased with depth." And changing the last sentence to be more explicit: "Furthermore, inorganic P decreased with depth more resources were invested to access it, supported by the consistently higher P targeting enzyme activity than N enzyme activity". (line 297 and line 302)

Lines 300-301: I would suggest adding another argument here at the end of this paragraph, to put P availability of your site/plots into perspective, by comparing it to other studies. Although you state that this is a both P and N poor site, your results indeed point to perhaps more inorganic P being cycled than organic P. Comparing with other studies could strengthen your discussion.

The end of the paragraph has been edited to strengthen the argument as suggested by the reviewer: "…microbes in the rhizosphere as an alternative to high energy cost enzyme production. Although soil P accumulates in the soil organic fraction with increasing soil age (Crews et al., 1995) this soil is also rich in metal oxides with large surfaces capable of adsorbing phosphate cations (Achat et al., 2016) which root activity in the rhizosphere can release with the help of organic acids without decomposing SOM (Adeleke et al., 2017)." (line 330)

Lines 308-310: Can you expand a bit more on how you can extrapolate your findings to turnover? I suggest bringing a bit the discussion from lines 337-339 (reference from Pineiro et al 2020) here as well.

The paragraph was amended to include: "Because we did not find a significant increase in potential enzyme activity in the rhizosphere (Table 6) this effect can instead be driven by microbial biomass turnover, community shift (Castañeda-Gómez et al., 2021) and a strong recycling of nutrients without large decomposition of SOM requiring enzyme activity. Although we can show that deep rhizosphere has an impact on available nutrients our study cannot assess if plants are utilising the increased availability, though increased root turnover (Piñeiro et al., 2020) has been reported suggesting that is the case." (line 340)

Lines 346-347: I suggest that this reference

https://www.sciencedirect.com/science/article/abs/pii/0016706181900240 could strengthen your argument.

We agree that the McGill and Cole (1981) reference is a good addition to strengthen the argument. The sentence has been edited to the following: "Mineral adsorbed P forms are however sensitive to root derived changes in pH (Jones and Darrah, 1994), representing a different mechanism for affecting the P cycle separate from SOM decomposition (McGill and Cole, 1981)." (line 382)

Lines 378-379: It could be useful to add a bit of the short-term versus long-term responses, as perhaps, the system might not be able to keep this faster cycling for too long under nutrient limitation.

The short term versus long term effect is briefly discussed in the section starting on line 376. One additional point was added to increase the contextual importance of new C on soil C stocks at the end of that same paragraph: "Tough a recent meta-analysis assigning short- and long-term effect of newly fixated C on soil C stocks could show that any short-term gains of C into SOM was gone after one to four years (van Groenigen et al., 2017).". (line 388)

**Technical corrections**

Line 38: Reference style should be revised.

The reference style was corrected throughout the manuscript. (line 38)

Line 47: "thus promote" should be "thus promoting".

The sentence has been changes to "thus promoting" as suggested by reviewer #1. (line 48)

Line 74: Add hyphen: depth-dependent.

Hyphen was added as suggested. (line 75)

Line 124: Remove ; after Londonderry clay and perhaps add a parenthesis.

The ";" was removed and the sentence was changed to: "…clay layer called Londonderry clay (Atkinson, 1988) found…". (line 128)

Lines 135-137: I suggest to revise this sentence to: "Although the depth of the transition layer differed throughout the site, the chemical properties are assumed to be similar within this zone across the plots, as the water periodically builds up above the clay before it drains, creating conditions for podzolification." (line 139)

The sentence was changed as suggested by reviewer 1. (line 139)

Line 168: From "mineralization, rate" to mineralization rate,"

The comma was removed as suggested by reviewer 1. (line 172)

Line 172: Should read "added in duplicate to fresh and…"

The sentence was changes as suggested by reviewer 1. (line 176)

Line 185: Shouldn't it be the effect of eCO2 and depth on roots, and not the other way around?

Sentence was changed as suggested by reviewer 1. (line 203)

Line 192: "analysis all CO2" should read "analysis of all CO2".

Sentence was changed as suggested by reviewer 1. (line 211)

Line 198-199: Reverse the results for better flow. Since you report a decrease with depth, state the 0-10cm results first, followed by the deeper layers.

Section was edited to reflect the comments of reviewer 1 as follows: "Fine root biomass density significantly decreased with depth and ranged from 0.12 mg·g-1 in the 0-10 cm depth to 2.75 mg·g-1 in the transition depth (Figure 2)." (line 217)

Line 205: Was the 24% increase for both 0-10 and 10-30 cm together (averaged) or the magnitude of change was the same for both depths separately?

The 24 % is referring to the magnitude change between bulk and rhizosphere soil as an average of the 0-10 cm and 10-30 cm depth (soil type as single factor). The sentence was edited to clarify this in the following way: "The DOC was significantly higher (by 24 %) in rhizosphere soil than bulk soil (Figure 2 and Table 1) when averaged across depth (0-10 and 10-30 cm depths)." (line 224)

Line 614: Remove the italics format.

The italics format was removed as suggested. (line 721)

Lines 221-222: Could you point to where (table, figure) we could see those results?

Figure reference were added as suggested by reviewer: "…ambient 10-30 cm rhizosphere (Figure 5), though…" (line 241)

Line 680: Parenthesis missing after 10-30 cm.

Parenthesis was added as suggested by reviewer. (line 798)

Line 627: Instead of "for of a mature" it should read "for a mature".

Edited as suggested by reviewer. (line 733)

Line 323: Initial caps missing in "rather the…".

Typo corrected. (line 359)

Line 326; PO4+ or PO43-?

The correct compound here is: "$PO_4^{-3}$", which has been corrected in text. (line 362)

**Dear Reviewer #2**

We thank reviewer #2 for carefully reading and suggesting edits to the manuscript titled "The influence of elevated $CO_2$ and soil depth on rhizosphere activity and nutrient availability in a mature Eucalyptus woodland" by Johanna Pihlblad et al., Biogeosciences Discuss (bg-2022-145) and are confident the work is better for it.

Below we have outlined how the suggested edits have been changed in response to the individual comments and questions.

Kind regards on behalf of all listed authors,

Dr Johanna Pihlblad

**Specific Comments**

Ln 17-18: I think this is one of the more interesting aspects of the study, however, this sentence in the abstract is very vague, and doesn't really tell me how eCO2 influenced nutrient availability. A stronger sentence would help the abstract.

The sentence in the abstract was changed to: "We found decreasing nutrient availability and gross N mineralization with depth, however this depth associated decreased was reduced under elevated $CO_2$ which we suggest is due to enhanced root influence." (line 17)

Ln 33 -34: The sentence structure here is rather awkward, can it be re-written for clarity?

The sentence was reworked to improve readability as follows: "Higher root exudation rates, stimulation of root growth and fine root production and turnover are all mechanisms that can potentially elicit SOM decomposition and subsequent nutrient release in the rhizosphere (Bernard et al., 2022)." (line 34)

Ln 38: Referencing of these two Iverson pprs needs to be fixed.

The referencing style has been corrected throughout the manuscript. (line 38)

Ln 47: 'thus promote' is a little awkward too - 'and thus promote' might work, or 'thus promoting', maybe?

The sentence has been changes to "thus promoting" as suggested by both reviewer #1 and reviewer #2. (line 48)

Ln 90: Is there any information on belowground activity under eCO2? Could water also be limited at this site? Also, what is the average rooting depth for these trees? How much for that is below 10 cm?

Yes, these questions are very interesting and warranted and the belowground activity is introduced later in this paragraph including the most up to date studies from the EucFACE facility. It is also common for Eucalyptus trees to have very deep structural and water seeking roots that can reach as far as down as 10-28 meters to access groundwater aquifers. No studies have been published from this specific site as of yet about the maximum rooting depth of this tree stand though long-term sensors at EucFACE have observed two aquifers in the top four meter of soil believed to be used by the trees (personal communication, Belinda Medlyn). Especially during periods of drought there can be a case where the top one meter of soil is very dry to the point of being hydrophobic, but the trees are not drought stressed due to access to groundwater. However, this isn't included here due to the lack of published studies supporting any statement on maximum rooting depth and if the trees are water limited or not. The paragraph was however amended to include a general statement about rooting depth for Eucalyptus trees as follows: "Additionally, Eucalyptus trees are known to have very deep roots to access water from groundwater aquifers (Laclau et al., 2013), though fine roots capable of nutrient acquisition are thought to be most abundant in the surface soil layers (Piñeiro et al., 2020)." (line 100)

Ln 143: I thought enzyme measurements were traditionally done on fresh soil samples. Why were these performed on (presumably)thawed samples, and how slowly were they thawed? Thawing too quickly will likely impose selection for tolerant members of the community. Maybe it's not a problem if all the samples are treated the same (enzymes measurements are potentials after all).

See response to reviewer #1 on similar comment. Too reply further, all the soils were indeed treated the same and the small amount of soil used meant that thawing happened quite rapidly after they were retrieved from the freezer which could impact the abundance of enzymes selectively. Because every plate did have a dried soil standard sample included, mainly to assess inter-plate variation, it allowed us to assess the expected range of an air-dried sample vs the frozen duplicates. No meaningful difference outside of the expected variability was observed, though the trend for the air-dried sample to have lower activity was noted (data not shown). Also, of note here is that the sample size for this comparison was small.

The specific actions taken in the manuscript are as follows: the enzyme data table and methodology were moved from the supplement to the main body of text as suggested by reviewer #1. Further, the caveats for the use of frozen soil was introduced in the method section 2.6 stating: "…however storage in -20 ˚C may have altered the potential enzymatic activity and comparisons with activities in fresh soil from other land-uses should be made with caution (Lane et al., 2022)." (section 2.6)

Ln 200: comma after 'where'.

Changed as suggested by reviewer #2. (line 219)

Ln 264: I'm not sure I quite understand the connection between microbial P and plant roots here. Maybe I missed the broader point, but I found this a little unclear.

> The connection is that both microbial P and fine root density both declined in response to $eCO_2$ in the 0-10 cm depth. The sentence was edited to improve this as follows: "Contrary to the non-response of the microbial C and N concentrations, the microbial P concentration decreased under $eCO_2$ in the 0-10 cm depth in the bulk soil (Figure 6C), this is similar to the negative effect of $CO_2$ on fine root density (Figure 2), suggesting that root density and microbial P respond similarly to $eCO_2$ since both decreased". (line 291)

Ln 271: What is the average water table depth at this site? I assume there is little strong redox chemistry occurring here that might impact the N-cycle and favor N-loss.

> Yes, we agree that the redox reactions likely found here is affecting the nutrient availability. As for the water table it was not found in the top ~1 meter that was investigated in this study. There are at least two groundwater aquifers found withing the first four meters of soil at this location (Personal communication with Belinda Medlin). The clay layer found at variable depths at the site (between 40 cm and 90 cm soil depth) appears to be semi-permeable allowing for occasional pooling of water above it before draining towards a drainage ditch or moving down deeper through the soil profile, which would indicate that changing redox conditions are occurring. Unfortunately, this was not specifically investigated in this study, being a one timepoint study in relatively dry conditions for this site, but we agree that there is likely a legacy effect on the nutrient cycling at this soil depth.

Ln 287: I'm not sure I'm convinced by this argument. The site is not limited by N or C, right? And the allocation to enzymes is trivial relative to that required to build microbial biomass (which increases under eCO2). I guess this interpretation also depends on how you interpret the enzyme data, which can be notoriously difficult. Does an increase in enzymes represent the availability of a given substrate (feast mode), or microbial limitation by a given substrate (bet-hedging approach). How you interpret your enzyme data goes some way to how you interpret the enzyme response.

> We have edited the context of the arguments weight by changing the sentence to include: "...enzymes (Olander and Vitousek, 2000), although there is no indication N or C are limiting for enzyme production in this system." (line 316)

Ln 335: I tend to think it means the 'potential' is there to decompose plant material down the soil column.

> The sentence has been edited to include a suggestion from reviewer 1: "… that plant matter have the potential of being decomposed throughout the soil profile." (line 371)